# Role of PDE4 Family in Cardiomyocyte Physiology and Heart Failure

**DOI:** 10.3390/cells14060460

**Published:** 2025-03-20

**Authors:** Ivan Sherstnev, Aleksandra Judina, Giovanni Battista Luciani, Alessandra Ghigo, Emilio Hirsch, Julia Gorelik

**Affiliations:** 1Cardiac Section, National Heart and Lung Institute (NHLI), Faculty of Medicine, Imperial College London, Hammersmith Campus, Du Cane Road, London W12 0NN, UK; ivan.sherstnev_02@univr.it (I.S.); a.judina18@imperial.ac.uk (A.J.); 2Department of Surgery, Dentistry, Pediatrics and Gynecology, Division of Cardiac Surgery, University of Verona, 37126 Verona, Italy; giovanni.luciani@univr.it; 3Department of Molecular Biotechnology and Health Sciences, Molecular Biotechnology Center “Guido Tarone”, University of Torino, 10126 Torino, Italy; alessandra.ghigo@unito.it (A.G.); emilio.hirsch@unito.it (E.H.)

**Keywords:** Phosphodiesterase 4, cAMP signalling, cardiomyocyte, β-adrenergic signalling, arrhythmia, cAMP compartmentation, A-kinase-anchoring protein, Phosphodiesterase inhibitor

## Abstract

Phosphodiesterase 4 (PDE4) is a key regulator of cyclic adenosine monophosphate (cAMP) signalling in cardiomyocytes, controlling contractility, calcium handling, and hypertrophic responses. PDE4 provides spatial and temporal precision to cAMP signalling, particularly under β-adrenergic stimulation, through its compartmentalised activity in subcellular nanodomains, including the sarcoplasmic reticulum, plasma membrane and nuclear envelope. This review highlights the cardiac PDE4 isoforms PDE4A, PDE4B and PDE4D, focusing on their distinct localisation and contributions to cardiac physiology and pathophysiology, particularly in heart failure and arrhythmias. Although PDE4 plays a smaller role in overall cAMP hydrolysis in human hearts than in rodents, its compartmentalised function remains critical. Recent therapeutic advances have shifted from pan-PDE4 inhibitors to isoform-specific approaches to enhance efficacy while minimising systemic toxicity. We discuss the potential of selective PDE4 modulators, gene therapies and combination strategies in restoring cAMP compartmentation and preventing maladaptive cardiac remodelling. By integrating rodent and human studies, this review underscores the translational challenges and therapeutic opportunities surrounding PDE4, positioning it as both a key regulator of cardiac signalling and a promising target for heart failure therapies.

## 1. Introduction

### 1.1. PDE Superfamily

Mammalian cells express 21 PDE genes, grouped into 11 superfamilies, collectively generating over 100 isoforms [1,2]. These isoforms exhibit tissue-specific and subcellular distribution, including within cardiomyocytes, allowing them to regulate specific GPCR signalling pathways and their associated cellular functions [3,4,5].

cAMP is a key second messenger in cardiomyocytes, generated by adenylate cyclases following β-adrenergic receptor activation. cAMP regulates contractility, heart rate and relaxation via protein kinase A (PKA) and other effector proteins [6,7]. PKA is a heterotetramer consisting of two regulatory (PKA-R) and two catalytic (PKA-C) subunits [8]. The regulatory subunit exists in two major isoforms, PKA-RI and PKA-RII, which differ in their biochemical properties, subcellular localisation and regulatory roles [9,10]. PKA-RI is predominantly cytosolic and more sensitive to cAMP, while PKA-RII is anchored to specific subcellular sites via AKAPs, allowing localised, compartmentalised signalling [9,11,12].

cGMP, synthesised by guanylate cyclases in response to nitric oxide or natriuretic peptides, regulates vasodilation, myocardial relaxation and antihypertrophic signalling via protein kinase G. Its interplay with cAMP, mediated by PDEs, modulates cAMP hydrolysis, ensuring precise regulation of cardiac signalling [6,13].

Based on substrate specificity, PDEs are classified into three main categories, as presented in Table 1.

In the heart, eight PDE families have been identified: PDE1 [41], PDE2 [32], PDE3 [35], PDE4 [42], PDE5 [43], PDE8 [19], PDE9 [44] and PDE10 [45] (Table 2). Nevertheless, the heart expresses mRNA for all PDEs except PDE6, though the importance of this is yet to be discovered [46].

### 1.2. PDE4 Structure and Function

PDE4 has garnered significant attention over the past few decades due to its prominent involvement in cardiac physiology and pathology. PDE4 uniquely hydrolyses cAMP with high specificity, positioning it as a central regulator of cAMP compartmentation within cardiomyocytes [67]. This precise regulation is crucial for maintaining cardiac function under both physiological and pathological conditions. The distinct roles of PDE4 isoforms in modulating β-adrenergic signalling [68,69,70], calcium handling [71,72] and hypertrophic responses [73,74] have highlighted their potential as therapeutic targets. Consequently, understanding the structural and functional nuances of PDE4 isoforms is pivotal for revealing new strategies to counteract cardiomyocyte maladaptive remodelling and develop innovative heart failure therapies.

The mammalian genome encodes four PDE4 genes—*PDE4A*, *PDE4B*, *PDE4C*, and *PDE4D* [14,15]—which, through alternative translational sites and splicing, yields approximately 25 distinct isoforms. Figure 1 illustrates the structural organisation, classification, regulation and enzymatic activity of PDE4 long and short isoforms in the heart and cardiomyocytes. Each isoform is potentially associated with specialised functions linked to specific subcellular localisations [3].

Based on the presence or absence of upstream conserved region (UCR) domains, the isoforms are classified as long, short or supershort forms [5]. The UCR domains, UCR1 and UCR2, are separated by the Linker Region 1 (LR1) and are crucial for the dimerisation of the enzyme [75,76]. Isoforms lacking UCR1—namely, the short and supershort forms—predominantly exist as monomers. The activity of PDE4 is regulated via the phosphorylation of Ser190 (for PDE4D) and Ser133 (for PDE4B) by PKA at the UCR1 domain [75,77,78,79,80,81], which enables PDE4 dimerisation, enhancing long isoform hydrolytic activity. The mitogen-activated protein kinase 1 (MAPK1, also known as ERK2) mediates phosphorylation at the C-terminal region, which causes an inhibitory effect on the long and supershort isoforms and activation of short isoforms [82,83,84,85,86]. This structural complexity underpins PDE4’s diverse roles in cardiac tissues, as detailed in the following sections.

Structural insights into the regulation of PDE4 activity, especially PDE4B, highlight the role of its long isoforms, which contain the upstream conserved regions (UCR1 and UCR2). These domains mediate dimerisation and allosteric regulation, distinguishing long PDE4 variants from short and supershort forms. Using X-ray crystallography and biochemical approaches, the authors reveal that UCR2 of one subunit interacts with the catalytic domain of the opposite subunit, thereby modulating enzymatic activity. This structural arrangement explains the differential regulatory properties among PDE4 isoforms and offers a mechanistic understanding of PDE4 inhibition [75,87]. A 3D structural model (Figure 2) of human PDE4B was created to visualise this organisation based on domain annotations adapted from [75]. This model corresponds to the long isoform of PDE4B, as shown in Figure 1, which includes UCR1 and UCR2 domains involved in dimerisation and regulation.

## 2. PDE4 Expression and Function in the Heart: From Species Variability to Subcellular Dynamics

### 2.1. PDE4 Expression and Function in Cardiac Tissue Across Different Species

While PDE4 localisation and functionality are conserved across rodent and human cardiac tissues [47,71,91,92,93], significant differences exist in their contributions to overall cAMP-PDE activity. In the human heart, PDE4 accounts for approximately 10% of total cAMP-phosphodiesterase activity, markedly lower than the 40–60% observed in rat and mouse hearts [47,72,93,94]. Despite these quantitative differences, PDE4 is similarly localised to the Z-band in cardiomyocytes in both rodents and humans [47]. Moreover, in humans, PDE4 is tethered to macromolecular complexes that also involve the same enzymes as in rodent hearts, including the PLN/SERCA2 complex and β_1_AR complexes [47].

Functionally, PDE4 regulates β-AR stimulated LTCC activity, a role conserved across species [71,91,92]. It also modulates RyR2 phosphorylation mediated by PKA, contributing to the fine-tuning of calcium cycling and excitation–contraction coupling [93]. This conservation underscores the critical role of PDE4 in localised cAMP signalling in cardiomyocytes, while species-specific differences in PDE4’s contribution to total cAMP-PDE activity highlight the importance of careful translation of findings from rodent models to human cardiac physiology.

Thus, the differences in PDE4 activity are primarily explained by the increased activity of other PDEs in humans, in particular, PDE3A [63,95,96]. The distinct kinetic properties of these enzymes further explain their differential regulation. It was revealed that PDE3 has a Km value roughly 10 times lower than PDE4 in the ventricular tissue of humans and guinea pigs, indicating that PDE3 is more efficient at hydrolysing cAMP at lower concentrations, while PDE4 becomes more active when cAMP levels rise, such as during β-adrenergic stimulation [97,98]. Similarly, PDE8 exhibits an even lower Km than PDE3 and is also expressed in cardiac tissues [53,54,99,100,101], suggesting a complementary role in maintaining basal cAMP levels. This suggests that PDE3 and PDE8 primarily regulate cAMP levels under basal conditions, while PDE4 plays a more prominent role when cAMP concentrations rise, such as during β-adrenergic stimulation. This kinetic disparity implies that PDE4’s regulatory role becomes more significant at elevated intracellular cAMP concentrations, matching its higher Km value, while PDE3 maintains functional activity at lower cAMP levels. It is also interesting that when PDE3 is inhibited, PDE4 becomes more active and reduces the impact of catecholamines on cAMP and L-type Calcium currents (ICa,L) [71]. The variations in enzyme properties help explain differences in how specific PDE4 isoforms are expressed and how they function in different species, as described below.

In murine cardiac tissues, PDE4A, PDE4B and PDE4D are expressed in similar proportions, while PDE4C is not detected [48]. Across species, including mice, rats and humans, specific PDE4 variants show distinct patterns of expression and localisation. All PDE4 isoforms were detected in mouse, rat and human hearts by Western blotting using pan-PDE4 antibodies. For example, the predominant isoform of PDE4A is PDE4A10 (105–110 kDa), with smaller amounts of PDE4A5 also present. In the PDE4B subfamily, PDE4B3 (~95 kDa) is the most abundantly expressed variant. The PDE4D subfamily exhibits the greatest diversity, with isoforms such as PDE4D3, PDE4D8 and PDE4D9 forming primary protein bands at 91–95 kDa, while PDE4D5 and PDE4D7 are detected as minor bands at higher molecular weights [47,102,103]. Thus, in cardiomyocytes, PDE4A5 [47], PDE4A10 [47,104], PDE4B3 [47,105], PDE4D3 [47,106], PDE4D5 [47,106,107,108], PDE4D7 [47], PDE4D8 [47] and PDE4D9 [47] were confirmed. Notably, all cardiac PDE4 splicing variants identified so far belong to long isoforms [109]. This observation highlights the critical role of PKA and β-adrenergic receptor signalling in regulating their function, as long isoforms are known to interact with specific macromolecular complexes to modulate cAMP levels effectively.

Cardiac PDE4 isoforms modulate cAMP levels across multiple subcellular compartments in cardiomyocytes [7]. The non-specific inhibition of PDE4 leads to a significant increase in cAMP during βAR activation in mice [7,48,110], rats [91,111] and humans [72]. In human atrial myocytes, PDE4, particularly the PDE4D subtype, plays a crucial role during β-adrenergic stimulation, where its inhibition significantly impacts calcium handling and contractile response, potentially contributing to arrhythmogenic events [72]. This modulation of cAMP and calcium handling highlights the broader role of PDE4 in heart function, as will be discussed next.

Key insights were provided about the species-specific effects of PDE4 inhibition with rolipram. In rat cardiomyocytes, PDE4 inhibition reduced cAMP-PDE activity, whereas its effects in guinea pig and human cardiomyocytes were negligible. Notably, in rats, rolipram had no effect under basal conditions but significantly enhanced contractility following β-adrenergic stimulation with isoproterenol or forskolin, increasing cardiomyocyte shortening by approximately 25% and 30%, respectively. These findings suggest that PDE4 inhibition primarily impacts cAMP signalling during β-adrenergic activation, which may be more pronounced under stress or disease states [94]. These findings were extended by investigating rolipram in failing human ventricular myocardium, where PDE4 inhibition failed to enhance inotropic or lusitropic responses, even with β-blocker treatment. While PDE4 inhibition appears less impactful in the healthy human heart, its role may still be relevant in pathological conditions where cAMP signalling is dysregulated. These results highlight a critical species difference, underscoring the need for caution when extrapolating findings from rodent models to human physiology. However, it is important to note that differences may also influence the discrepancies between these studies in experimental models (e.g., isolated cardiomyocytes vs. whole myocardial tissue), conditions (e.g., β-adrenergic activation) and disease states. While PDE4 inhibition appears less impactful under baseline conditions in the human heart, its role may still be relevant in pathological states or during heightened cAMP signalling [112].

### 2.2. Regional Differences in PDE4 Activity

βARs play a central role in regulating cAMP signalling, with PDE4 acting as a key modulator to ensure precise compartmentalisation and control of downstream cardiac responses [105,113,114]. Figure 3 illustrates the intricate network of βAR-cAMP-PDE4 signalling, highlighting PDE4’s role in shaping distinct cAMP pools that regulate various cardiac functions, including calcium handling, contractility and gene expression. Regional variations in PDE4 activity within the heart have significant implications for cardiac functions. Professor Gorelik’s research group reported that intra-chamber variability in PDE4 activity within the left ventricle limits β_2_AR-associated cAMP diffusion in basal but not apical cells. Reduced PDE4 enables higher cAMP diffusion to the PKA-RII domain, leading to enhanced contractile response [114]. These intra-chamber differences also extend to variations between the left and right ventricles, as discussed below.

Enhanced contractile response of right ventricular cardiomyocytes to catecholamine stimulation is attributed to elevated PDE3 and PDE4 activities in left ventricular cardiomyocytes [115]. This phenomenon may represent an adaptive mechanism to meet the increased oxygen demand during heightened physical activity by enhancing pulmonary circulation [116]. Specifically, canine models have shown a greater right ventricular cardiomyocyte response to βAR stimulation compared to left ventricular cells, associated with higher PDE3 and PDE4 activities in the left ventricle [115]. Beyond chamber-specific differences, PDE4 activity also varies within specific regions of the ventricular wall.

Cardiac chambers display distinct structural and functional characteristics, which may influence PDE4 activity. For example, a significant difference was demonstrated in calcium signalling between atrial and ventricular cardiomyocytes, linked to structural variations like the presence of T-tubules in ventricular cells [117]. The remodelling of right ventricular cardiomyocytes under conditions such as pulmonary hypertension was also reported, which could alter cAMP-PDE compartmentation [118].

Within the LV, apex-to-base differences in β_2_AR signalling have been identified. Despite similar cytosolic cAMP levels, it was shown that apical cardiomyocytes exhibit greater contractility in response to β_2_AR stimulation than basal cells. This effect was attributed to enhanced membrane organisation and higher PDE4 activity in basal cardiomyocytes, which restricts cAMP diffusion and modulates β_2_AR signalling [114]. Subcellular compartmentation is critical in β_2_AR signalling, mediated by PDE4 and other proteins. It was demonstrated that β_2_AR localises predominantly to caveolae, invaginations of the sarcolemma enriched in caveolin-3 [119]. Basal cardiomyocytes, which have a higher density of caveolae, exhibit tighter control over cAMP signalling than apical cells [120]. This organisation enables precise spatial regulation of cAMP, contributing to regional functional differences [114]. Moreover, studies using FRET-based sensors showed that PDE4 activity preferentially restricts β_2_AR-associated cAMP signalling to specific microdomains, such as the nucleus, further emphasising the importance of subcellular localisation in PDE4 function [7,114,118,121]. Finally, differences in PDE4 activity across species, including humans and rodents, may reflect evolutionary adaptations to distinct cardiac demands. Such variations underscore the need to carefully interpret animal studies when exploring PDE4 as a human therapeutic target.

### 2.3. PDE4 Isoform-Specific Functions and Subcellular Localisation

PDE4 is the primary regulator of localised cAMP signalling in cardiomyocytes [6,13,46]. Each PDE4 isoform operates in different subcellular microdomains, producing specific effects on cAMP behaviour and subsequent signalling [6,46,114,118]. The unique roles of these isoforms present potential therapeutic targets, especially regarding cardiac physiology.

Below, the specific subcellular localisation of cAMP by PDE4 will be discussed, including its role in β-adrenergic signalling and calcium current regulation, its role in the sarcoplasmic reticulum and signalling at the nuclear envelope. Each of these compartments provides a unique environment where PDE4 finetunes cAMP dynamics to maintain cardiac function.

#### 2.3.1. Plasma Membrane: Modulation of βAR Signalling

At the plasma membrane, various splicing variants of PDE4D interact with β_1_AR and β_2_AR, either directly or via β-arrestin. These interactions modulate the receptors’ sensitivity to ligand stimulation and regulate downstream signalling pathways, including those mediated by other G-protein-coupled receptors (GPCRs). For example, the ability of β-arrestin to scaffold PDE4D ensures that localised cAMP signalling is tightly regulated, which can indirectly influence the activity of other GPCRs sharing overlapping signalling pathways [68,70,72]. This mechanism highlights the importance of PDE4D in maintaining specificity within complex cAMP signalling networks.

In neonatal mouse cardiomyocytes, PDE4D demonstrates isoform-specific regulation of β_2_AR signalling. Notably, PDE4D selectively regulates β_2_AR without affecting β_1_AR signalling, emphasising its role in localising and fine-tuning cAMP signals [122]. It was also shown that PDE4A plays a secondary role in the regulation of βAR signalling in neonatal mouse cardiomyocytes [122]. In PDE4A-KO mice myocytes, isoproterenol-stimulated contraction remains similar to wild types; however, residual PDE4 activity inhibition with rolipram enhances contraction [122].

Ontogenetic studies on the chick ventricular myocardium further support the role of PDE4 in developmental β-adrenergic modulation. Early-stage chick hearts primarily express a Ca^2^⁺/calmodulin-sensitive PDE with characteristics consistent with PDE1C [123], which regulates cAMP and cGMP signalling in human hearts [59]. However, at the late embryonic stage (18E), an additional PDE activity peak emerges, resembling PDE4 based on its substrate specificity and elution profile [123]. This transition coincides with a well-documented 10-fold reduction in β-adrenergic sensitivity of cardiac contraction between embryonic day 16 and hatching (21E), a period marked by the onset of adrenergic neuroeffector transmission in the right ventricle [124]. The emergence of PDE4 at this stage suggests a protective role in limiting excessive β-adrenergic stimulation as sympathetic innervation begins, mirroring its function in mature mammalian hearts, where it prevents cAMP overaccumulation and downstream hyperactivation.

Furthermore, the recruitment of PDE4D5 to β_2_AR by β-arrestin is critical for attenuating pro-hypertrophic signalling pathways. This process involves the inhibition of exchange protein directly activated by cAMP 1 (EPAC1) and calcium/calmodulin-dependent protein kinase II (CaMKII), key mediators of β_2_AR-induced hypertrophic responses [125].

PDE4B and PDE4D isoforms regulate cAMP at the caveolin-rich membranes of rodent cardiomyocytes, with PDE4D predominantly associated with β_2_AR signalling and PDE4B with β_1_AR signalling [48,105,114,126]. PDE4B plays a critical role in modulating cAMP levels near Ryanodine Receptor 2 (RyR2) and negatively regulates LTCC activity, thereby reducing βAR-stimulated ICa,L [48,127]. The loss of PDE4B in neonatal mouse cardiomyocytes results in elevated cAMP at the sarcolemma, enhancing the activity of key proteins like CaV1.2 and RyR2, which are essential for cardiac contraction [105]. In contrast, PDE4D activity is more closely linked to cAMP regulation at the SERCA2a and PLN, influencing calcium cycling [126,127]. Unlike PDE4B and PDE4D, PDE4A does not associate with the LTCC and has no significant impact on β-AR-stimulated ICa,L. Patch-clamp studies show no changes in ICa,L potentiation compared to wild-type mice, suggesting that PDE4A does not directly regulate cardiac calcium signalling [48]. Consistent with this, in PDE4b–/– mice, but not in PDE4a–/– mice, the β-AR response of ICa,L increased, along with an enhancement in cell contraction and Ca^2^⁺ transients, further supporting the specific role of PDE4B in modulating ICa,L and β-AR signalling [48]. The precise localisation of these isoforms underscores their distinct and complementary roles in cardiomyocyte function.

#### 2.3.2. Calcium Current Regulation by PDE4

PDE4 enzymes play a crucial role in modulating calcium dynamics and influencing the susceptibility to arrhythmias in cardiomyocytes [71,72,93]. One research revealed that the inhibition of PDE4 elevates cAMP and enhances calcium currents in human atrial myocytes, leading to increased spontaneous calcium release and a heightened risk of arrhythmias during β-adrenergic stimulation [72]. They also identified that PDE4 activity diminishes with advancing age and experiences a further decline in patients with AF, thereby linking PDE4 dysfunction to increased arrhythmic vulnerability. Additionally, it was demonstrated that the regulatory function of PDE4 becomes particularly significant under conditions of elevated cAMP in human and rabbit atrial cardiomyocytes, notably during β-adrenergic stimulation or the inhibition of PDE3, thereby highlighting its essential role in maintaining calcium homeostasis during periods of physiological stress [71].

The crucial role of PDE4 in regulating the cAMP signalling pathway within adult rat ventricular myocytes was discovered. PDE4 primarily modulates cAMP signalling associated with Gαs-coupled glutamate receptors (Glu-R), which enhances the β_1_AR-mediated increase in L-type Calcium current (ICa,L). Specifically, PDE4 amplifies the Glu-R-induced elevation in cAMP levels, leading to greater stimulation of ICa,L, thereby potentiating β_1_AR responses. While both PDE4 and PDE3 collaborate to regulate β_1_AR and β_2_AR responses, PDE4 becomes the dominant regulator when PDE3 activity is reduced. Furthermore, PDE4 restricts the diffusion of cAMP generated by prostaglandin E1 receptor (PGE1-R) signalling, although this pathway does not significantly influence ICa,L [128].

The importance of PDE4 in calcium regulation is further highlighted by its efficacy in modulating ICa. It was identified that PDE3 and PDE4 are the predominant PDEs involved in regulating basal ICa. By utilising selective PDE inhibitors in isolated rat ventricular myocytes, they revealed that PDE4 plays a dominant role in controlling cAMP levels near LTCC. Upon stimulation, which increases cAMP production, all four PDE subtypes contribute to the calcium current response, exhibiting a clear hierarchy of potency: PDE4 > PDE3 > PDE2 > PDE1 [129]. This hierarchical regulation indicates the critical role of PDE4 in adjusting cellular calcium dynamics in response to diverse signalling inputs.

In experiments with PDE4D−/− mice, increased contractility was observed at baseline and during β-adrenergic stimulation, accompanied by enhanced calcium transients and SR calcium content without changes in ICa,L [130]. These results demonstrate that PDE4D specifically modulates SR calcium cycling, ensuring balanced calcium release and reuptake. Collectively, these findings highlight PDE4’s critical function in protecting against calcium dysregulation, particularly under conditions of stress or disease.

#### 2.3.3. Sarcoplasmic Reticulum: Local Regulation of cAMP Microdomains

PDE4 plays a critical role in regulating calcium dynamics within the sarcoplasmic reticulum (SR). It was demonstrated that inhibiting PDE4 with Ro 20-1724 enhances calcium loading in the SR while simultaneously promoting pro-arrhythmic calcium leaks via PKA and calcium/calmodulin-dependent protein kinase II (CaMKII) pathways in isolated rat cardiomyocytes. Interestingly, the pro-arrhythmic risk could be mitigated through CaMKII inhibition, which preserved positive inotropic effects [113]. These findings underscore the importance of precisely modulating PDE4 activity to balance inotropic benefits against arrhythmogenic risks.

Within the SR, PDE4D is associated with the SERCA2–PLN complex, where it plays a key role in controlling cAMP levels. Reduced PDE4 activity near SERCA2, as observed in mouse hypertrophied cardiomyocytes, can enhance calcium reuptake by increasing PLN phosphorylation, offering potential compensatory benefits in heart failure [131,132,133,134]. However, the implications of PDE4’s role in SR cAMP microdomains are context-dependent, with both beneficial and detrimental effects observed depending on the target and pathological state.

PDE4D3, a specific isoform of PDE4D, is localised within the RyR2 macromolecular complex in the SR. In human heart failure, the association between PDE4D3 and RyR2 is diminished, contributing to hyperphosphorylated and ‘leaky’ RyR2 channels [64]. This pathogenic mechanism is further highlighted by studies in PDE4D-deficient mice, where PKA hyperphosphorylation of RyR2 leads to increased susceptibility to exercise-induced arrhythmias and late-onset dilated cardiomyopathy [93]. Recent investigations using transgenic mice expressing RyR2-targeted cAMP biosensors revealed reduced RyR2-associated PDE4 in hypertrophic conditions, resulting in elevated RyR2 phosphorylation in response to β_2_AR stimulation [73].

#### 2.3.4. Nuclear Envelope

PDE4 enzymes not only regulate calcium dynamics but also play a pivotal role in localised cAMP signalling at the nuclear envelope, where more than half of total PDE4 hydrolytic activity in adult cardiomyocytes occurs [135]. This nuclear compartmentalisation highlights the enzyme’s involvement in transcriptional regulation and hypertrophic signalling pathways. In cardiomyocytes of PDE4d−/− mice, the absence of PDE4D enhances nuclear PKA responses to βAR stimulation, contributing to late-onset dilated cardiomyopathy [136]. These findings link nuclear PDE4D activity to the prevention of pathological cAMP-mediated responses under stress.

Mechanistically, the overexpression of the C-terminal segment of the UCR1 domain (UCR1C) in long-PDE4 isoforms has been shown to inhibit nuclear PKA activity, reducing phosphorylation of the cAMP-responsive element-binding protein (CREB) transcription factor. This suppression of CREB phosphorylation mitigates cardiomyocyte hypertrophy [137]. Furthermore, PDE4D regulates a specific nuclear cAMP pool responsible for PKA-mediated phosphorylation of HSP20, a protein whose phosphorylation provides cardioprotective effects against hypertrophic stimuli [74]. Collectively, these studies illustrate how nuclear PDE4 activity integrates with broader cAMP signalling to protect cardiomyocytes from maladaptive hypertrophic responses.

## 3. A-Kinase-Anchoring Proteins in Cardiomyocytes: Regulators of cAMP Compartmentation and Signalling

### 3.1. Overview of AKAPs: Precision in PKA Signalling

AKAPs are a family of structurally diverse proteins that spatially organise cAMP-dependent PKA signalling activity. Most AKAPs tether the type II PKA holoenzyme to specific subcellular structures through their conserved RII-binding domains [138]. However, it has been shown that AKAP1 can also anchor type I PKA, demonstrating functional diversity among AKAPs [139,140]. AKAPs contain unique targeting domains that localise the AKAP/PKA complex to specific intracellular sites, enabling precise phosphorylation of nearby substrates and maintenance of cAMP/PKA signalling specificity and efficiency [138].

The nomenclature of AKAPs remains complex, as historically, multiple names may refer to the same AKAP discovered by multiple groups. This is summarised in Table 3, which presents the various aliases, key functions and supporting references for each AKAP.

Conventional AKAPs regulate PKA signalling in distinct cardiomyocyte compartments, demonstrating their importance in maintaining compartmentalised cAMP/PKA responses essential for cardiac function [144,145,150,155,174]. However, recent studies have identified non-conventional AKAPs that expand the functional repertoire of anchoring proteins, introducing additional layers of complexity to cardiac signalling.

### 3.2. Non-Conventional AKAPs in Cardiac Signalling

Non-conventional AKAPs are a group of anchoring proteins that go beyond the classical role of tethering PKA to specific subcellular locations. Unlike classical AKAPs, non-conventional AKAPs, such as phosphoinositide 3-kinase gamma (PI3Kγ) and talin, provide dynamic scaffolding structures to their cellular environments, thereby promoting increased regulatory complexity in signalling events [175,176,177,178,179].

PI3Kγ, a class IB PI3K, acts both as a kinase, catalysing the production of phosphatidylinositol (3,4,5)-trisphosphate (PIP3), and as a scaffold that anchors PKA and PDEs to specific signalling microdomains [176,177,178]. Anchored PKA activates PDE to enhance cAMP degradation and phosphorylates p110γ to inhibit PIP3 production [175]. Studies using kinase-dead and -deficient mouse models have revealed distinct roles for PI3Kγ. In kinase-dead mice, the ability to regulate cAMP in specific cellular regions is preserved because the scaffold function of PI3Kγ is intact, while its kinase activity is absent. In contrast, kinase-deficient mice lack both scaffold and kinase functions and display excessive cAMP accumulation and aberrant β-adrenergic signalling, leading to disrupted calcium handling and contractility [177,178]. Thus, PI3Kγ plays a dual role in cardiomyocyte β-adrenergic signalling, where it regulates cAMP dynamics through its scaffold function while influencing downstream pathways, such as Akt signalling, through its kinase activity [180,181].

Talin, a focal adhesion protein, represents another non-conventional AKAP with its function tightly linked to mechanotransduction [182,183]. Talin bridges integrins and the actin cytoskeleton, forming a mechanically sensitive scaffold that responds to cellular tension [182]. Recent studies have demonstrated that talin binds PKA in a force-dependent manner, working as a mechanically gated AKAP [179]. Under mechanical stress, talin’s R9 domain unfolds, exposing a cryptic PKA-binding site. This interaction anchors PKA to focal adhesions and enables the phosphorylation of targets such as Vasodilator-Stimulated Phosphoprotein, a key regulator of actin dynamics. The talin–PKA complex illustrates a new paradigm of signal regulation, where mechanotransduction directly influences the spatial and temporal activity of PKA [179].

Non-conventional AKAPs, such as PI3Kγ and talin, extend the regulatory scope of cAMP/PKA signalling in cardiomyocytes [176,177,178,179]. PI3Kγ anchors PDE4 to βAR microdomains, regulating cAMP levels and calcium homeostasis [176,177,178]. While talin’s direct role in PDE4 regulation remains unstudied, its classification as an AKAP suggests potential involvement in cAMP signalling pathways [179].

PI3Kγ and talin underscore the importance of non-conventional AKAPs in fine-tuning cAMP signalling, ensuring calcium homeostasis, contractility and protection against cardiac remodelling.

### 3.3. AKAP-PDE4-PKA Complexes: Masters of cAMP Regulation

PDE4D3 emerges as a major regulator of localised cAMP dynamics through anchoring various AKAPs, including AKAP9, mAKAP, and AKAP12 [106,184,185,186].

Several works emphasise the importance of the complex between mAKAP and PDE4D3 [186]. mAKAP establishes a critical scaffold at the nuclear envelope of cardiomyocytes, orchestrating a tightly regulated cAMP signalling network essential for maintaining cardiac homeostasis. The basis of this network is PDE4D3, which binds to mAKAP and acts as a negative feedback regulator by degrading cAMP generated in response to adrenergic stimulation [186]. This localised cAMP control is enhanced by PKA, also anchored by mAKAP, which phosphorylates PDE4D3 at Ser-13 and Ser-54, strengthening its enzymatic activity and ensuring precise spatial and temporal modulation of cAMP signalling [187,188].

At the SR, mAKAP is directly associated with RyR through a conserved leucine zipper motif, forming a key signalling complex that regulates calcium dynamics [189]. This localisation facilitates PKA-mediated phosphorylation of RyR at Ser-2808, increasing its open probability and calcium sensitivity. However, in pathological states such as heart failure, reduced PDE4D3 levels in the mAKAP complex lead to RyR hyperphosphorylation, contributing to calcium leakage from the SR and promoting arrhythmias and cardiac dysfunction [93]. These findings illustrate how the mAKAP–PDE4D3 complex integrates cAMP and calcium signalling, influencing both nuclear and cytoplasmic processes, with critical implications for cardiomyocyte function and maladaptive remodelling.

Within cardiac myocytes, PDE4D3 associates with the IKs potassium channel complex via AKAP9, ensuring the precise modulation of channel activity under adrenergic stimulation [184]. Similarly, AKAP12 anchors PDE4D3 and PKA to subplasmalemmal domains, enabling the rapid degradation of cAMP and maintaining signalling accuracy at the plasma membrane [106]. Interestingly, AKAP12 anchors PDE8A in a similar way [53,100]. The disruption of both complexes results in dysregulated cAMP gradients, with prolonged signalling evident upon AKAP12 knockdown [106,184]. Altogether, these studies underscore the critical role of AKAP-anchored PDE4D3 in the organisation of cAMP compartmentation and supporting cellular homeostasis.

PI3Kγ, as a non-conventional AKAP, can form complexes with different PDEs [175,176,177]. PI3Kγ anchors PKA alongside PDE isoforms, particularly PDE4A, PDE4B and PDE3A in cardiomyocytes. These PI3Kγ-regulated PDEs reduce cAMP levels, thereby restricting PKA-mediated phosphorylation of the LTCC and PLN, ensuring the precise control of calcium handling and cardiac contractility [177]. In the absence of PI3Kγ, PDE4 activity is impaired, resulting in the hyperphosphorylation of calcium-handling proteins and arrhythmic calcium transients [177]. Furthermore, pharmacological studies have revealed that targeting the PKA-PDE4B/PDE4D complexes mediated by PI3Kγ could offer therapeutic strategies to elevate cAMP in chronic obstructive respiratory diseases [77,176].

AKAP–PDE4–PKA complexes are essential for compartmentalised cAMP signalling, enabling the precise regulation of calcium dynamics, contractility and membrane excitability in cardiomyocytes [93,106,184,185,186,189]. Examples such as the mAKAP-PDE4D3 scaffold at the nuclear envelope and SR and PI3Kγ–PDE complexes underscore the critical role of localised cAMP control in maintaining cardiac homeostasis [106,184,185,186]. Dysregulation of these complexes has been implicated in arrhythmias, heart failure and maladaptive remodelling, making them promising therapeutic targets [93,189]. However, further research is needed to fully understand their mechanisms and evaluate clinical applicability, paving the way for targeted therapies addressing cAMP-driven cardiac pathologies.

## 4. PDE4 and AKAPs in Heart Failure: Implications for cAMP Signalling and Cardiac Remodelling

### 4.1. PDE4 and Heart Failure

Heart failure and cardiac hypertrophy are characterised by significant alterations in the expression and activity of PDE4 in cardiomyocytes, which disrupts cAMP signalling and contributes to maladaptive remodelling. A decrease in the activity of PDE4A and PDE4D has been observed in failing hearts [47], while compartment-specific alterations in the activity of PDE4 and PDE3 have been reported in hypertrophic ventricular myocytes [190]. Chronic exposure to catecholamines leads to a reduction in PDE4 and PDE3 activity in PKA-RI compartments while simultaneously enhancing PDE4 activity in PKA-RII compartments, thereby exacerbating pathological remodelling. The observed shift in the equilibrium between PDE3 and PDE4 favours PDE3 in pathological conditions, diminishing the regulatory influence of PDE4 on cAMP levels [191,192]. PKA-RI is predominantly found in cytosolic regions and responds rapidly to transient cAMP increases. In contrast, PKA-RII is enriched in subcellular structures, including the sarcoplasmic reticulum and nuclear envelope, enabling sustained and localised cAMP signalling [126,193]. Interestingly, in a rat model of cardiac hypertrophy induced by thoracic aortic banding, the total cAMP hydrolytic activity and activities of PDE4 and PDE3 isoforms—including PDE4A and PDE4B—are reduced, whereas PDE4D activity remains unchanged [191]. These findings suggest that PDE4A may play a role in pathological remodelling during heart failure, potentially through its contribution to broader changes in cAMP signalling dynamics or the synergistic effect of cAMP inhibition.

Moreover, the redistribution of β_2_ARs and the alteration of cAMP compartmentation reduce the protective mechanisms afforded by β_2_ARs, resulting in desensitisation and compromised cardiac function [194]. The restoration of PDE4B activity through transgenic overexpression or adeno-associated virus serotype 9 (AAV9) gene therapy alleviates pathological remodelling and normalises cAMP signalling, highlighting the therapeutic potential of PDE4 targeting [110]. The overexpression of PDE4B3 reduces cardiac hypertrophy and fibrosis while restoring cAMP compartmentation in RyR and caveolin-rich microdomains. This therapy also decreases RyR2 phosphorylation and arrhythmic events, showing potential in patient-derived cardiomyocytes with specific genetic mutations. These findings suggest that PDE4B3 modulation could offer targeted treatment for heart failure and associated arrhythmias [194]. Furthermore, it was reported that the deletion of PDE4D leads to an increase in RyR2 phosphorylation and an elevated risk of arrhythmias, thereby further accelerating the progression of heart failure [93]. Altogether, these findings highlight the key role of PDE4 isoforms in the maintenance of cardiac function and their potential as therapeutic targets in the context of heart failure.

Targeting PDE4 isoforms presents a promising treatment strategy for heart failure by restoring cAMP compartmentation and calcium dynamics while mitigating the adverse effects of prolonged catecholamine stimulation. Isoform-specific modulation, such as increasing PDE4 activity in protective microdomains (e.g., SERCA2 and the nuclear envelope) or blocking pro-arrhythmic pathways driven by PKA and CaMKII, may tackle significant pathological mechanisms. Innovative strategies, such as gene therapy and pharmacological agents, present chances to utilise the therapeutic potential of PDE4 to diminish maladaptive cardiac remodelling and enhance results in heart failure.

### 4.2. AKAPs in Heart Failure: Mechanistic Roles and Pathological Impact

AKAPs are essential in the spatial and temporal regulation of signalling pathways critical for cardiac physiology and pathology [145,150,155,173,177]. AKAP1 exerts an anti-hypertrophic effect by sequestering calcineurin, thereby inhibiting NFAT-dependent transcriptional programs that can lead to pathological cardiomyocyte growth. The knockdown of AKAP121 leads to hypertrophy in neonatal rat cardiomyocytes, whereas its overexpression suppresses isoproterenol-induced hypertrophic responses, positioning it as a potential therapeutic target for calcineurin/NFAT modulation [195]. Similarly, AKAP13 acts as a scaffold for protein kinase D (PKD), coordinating the phosphorylation and nuclear export of HDAC5 to relieve MEF2 repression and promote hypertrophic gene expression. Disrupting the PKD binding site on AKAP13 decreases hypertrophic responses, emphasising its role in pathological remodelling [172].

The nuclear envelope-anchored AKAP6 integrates diverse hypertrophic signals, coordinating transcriptional regulation through interactions with PLCε [196], PKD [197] and various transcription factors, including MEF2 [197,198] and NFAT [197]. These complexes facilitate the precise control of local signalling, such as PKD-mediated HDAC phosphorylation, driving hypertrophic remodelling [197]. The loss of AKAP6, a scaffold that coordinates key signalling pathways at the nuclear envelope, reduces pressure-overload-induced hypertrophy, fibrosis and apoptosis, as shown by improved survival and attenuated remodelling in AKAP6-deficient mice [196,197,199]. Furthermore, alterations in AKAP-mediated cAMP dynamics are evident in heart failure models. For example, AKAP7 and PDE4 regulate β_2_AR signalling in the PLN/SERCA2a microdomain. Dysregulation of this axis in HFpEF due to obesity and diabetes leads to enhanced β_2_AR-mediated cAMP production, PLN phosphorylation and calcium reuptake, a compensatory mechanism that may contribute to maladaptive remodelling [200].

Together, these findings underscore the complexity and therapeutic promise of targeting AKAPs to modulate maladaptive cardiac remodelling and improve outcomes in heart failure.

## 5. Therapeutical Insights and Future Perspectives

Efforts to refine PDE4 inhibitors have been ongoing for decades, aiming to enhance selectivity, minimise side effects and expand their therapeutic potential [201,202,203,204,205,206]. These advancements have led to the development of several PDE4 inhibitors, including roflumilast, apremilast and crisaborole, which have completed Phase IV clinical trials. Additionally, other drugs—such as Hemay005, cilomilast and tanimilas—are in various stages of clinical development, with some still undergoing Phase III trials while others have progressed further for indications ranging from chronic respiratory diseases to inflammatory and dermatological conditions (Table 4) [202,207,208,209]. Further advancements in PDE4 inhibitor development have driven the need for greater specificity to enhance efficacy and reduce adverse effects. This has led to the emergence of targeted PDE4 inhibitors, such as GSK256066 (selective for PDE4B) [210] and BPN14770 (selective for PDE4D) [205], which are currently in phase II clinical trials for Chronic Obstructive Pulmonary Disease (COPD) and Fragile X Syndrome, respectively (Table 4). Currently, KIT2014, a PI3Kγ-mimetic peptide that inhibits PDE4B and PDE4D [176], is undergoing a phase I clinical trial as a potential alternative to classical PDE4 inhibitors (Table 4).

Roflumilast, a pan-PDE4 inhibitor approved in 2010 for severe chronic obstructive pulmonary disease, exerts its effects via its active metabolite, roflumilast-N-oxide [208]. Importantly, pooled clinical trial data have demonstrated that roflumilast does not increase cardiovascular risk, with rates of major adverse cardiovascular events (MACEs) remaining low and comparable to placebo [212]. This favourable safety profile is particularly relevant given the potential for PDE4 inhibitors to influence cardiovascular physiology. Similarly, apremilast, introduced in 2014, is a pan-PDE4 inhibitor initially approved for psoriasis and later for psoriatic arthritis and Behçet’s disease. Despite its lack of isoform specificity, apremilast has demonstrated significant efficacy and tolerability in treating inflammatory conditions, making it a valuable therapeutic option [209].

Recognising the need for greater isoform specificity, researchers have shifted their focus to developing inhibitors targeting specific PDE4 isoforms, particularly PDE4B and PDE4D, which hold promise in cardiac disease management. In the heart, localised distortions of PDE activity are implicated in several pathological conditions. For example, maladaptive remodelling in heart failure is associated with altered cAMP degradation within key microdomains, such as those around the SR and LTCC. Reduced cAMP levels and impaired PKA activity in these regions disrupt calcium handling and contractility, contributing to disease progression [47,93,189,190]. In arrhythmias, increased PDE activity within RyR2 microdomains leads to inadequate cAMP signalling, promoting calcium leaks and electrical instability [71,72,93]. These findings underscore the importance of precisely targeting specific PDE4 isoforms within distinct subcellular domains to restore localised cAMP dynamics without systemic disruption.

Novel PDE4D allosteric modulators were developed that selectively interact with the enzyme’s UCR2 regulatory domain. Unlike traditional active site inhibitors that fully suppress activity, these modulators partially inhibit enzymatic activity, preserving the spatial and temporal control of cAMP signalling. This selective inhibition reduces adverse effects, such as emesis, a common limitation of pan-PDE4 inhibitors. Co-crystal structural studies revealed how these modulators stabilise the UCR2 domain in a “closed” conformation, effectively capping the catalytic domain to modulate activity [206]. Similarly, PDE4B/D-selective inhibitors have shown improved selectivity and reduced toxicity [211,213], further supporting their potential in cardiac applications. Recent advances in PROteolysis-TArgeting Chimera (PROTAC) technology offer a novel approach to PDE4 modulation. For instance, the BI 1015550-based PROTAC, KTX207, selectively targets PDE4D shortforms with remarkable potency, achieving an IC50 of approximately 10 pM for degradation. This strategy enhances efficacy in suppressing inflammatory markers and promises a superior side effect profile compared to conventional catalytic site inhibitors, making it a promising tool for precise cAMP regulation in cardiac applications [214].

Targeting PDE4 isoforms within specific cardiac microdomains offers a promising therapeutic strategy for conditions like heart failure and arrhythmias. By restoring localised cAMP signalling, these approaches may address pathological remodelling and improve cardiac function while minimising systemic toxicity. Continued research into isoform-selective PDE4 inhibitors could pave the way for precision therapies that balance efficacy and safety in cardiovascular diseases.

Alternative future therapeutic strategies for cardiac diseases could involve combining isoform-specific PDE4 inhibitors with precision targeting of their interacting partners, such as AKAPs and POPDC1, to enhance localised cAMP regulation within key cardiac microdomains. This approach could help restore cAMP dynamics in pathological conditions like heart failure and arrhythmias, where disrupted compartmentalisation contributes to maladaptive remodelling and impaired contractility. Advances in gene therapy and mRNA-based technologies offer additional potential by enabling the precise modulation of specific PDE4 isoforms in subcellular domains such as the sarcoplasmic reticulum and plasma membrane. These strategies could directly address local cAMP deficiencies, reduce maladaptive remodelling and improve calcium handling and overall cardiac function. Furthermore, high-throughput screening platforms and computational models could aid in the discovery of novel allosteric modulators or small molecules, paving the way for safer and more effective PDE4-based therapies tailored to individual patient needs.

## 6. Conclusions: Can PDE4 Still Be Considered Only a Minor Helper in the Heart?

The PDE4 family is essential in regulating cardiomyocyte physiology and pathophysiology, intricately balancing localised cAMP signalling [6,7,67,71,72,93]. In the heart, the PDE4A, PDE4B and PDE4D isoforms function in distinct subcellular microdomains, affecting cardiac contractility, calcium handling and hypertrophic responses often via complex formation with other regulatory and effector proteins [48,105,114,126]. These complexes maintain homeostasis and critical regulators during normal physiological conditions, while their dysfunction can contribute to severe pathological states, including heart failure and arrhythmias [48,106,122,184].

Species-specific and regional variations in PDE4 activity underscore the complexity of translating rodent-based findings to human cardiology [47,72,93,94,112]. For instance, while PDE4 plays a minor role in overall cAMP hydrolysis in human hearts [94,112], it is crucial for compartmentalised signalling, particularly under β-adrenergic stimulation [68,69,70,114]. The cooperative dynamics between PDE4 and PDE3 further emphasise the necessity of a deep understanding of their roles in modulating cardiac responses, especially in the disease context.

Therapeutic strategies targeting PDE4 have evolved significantly, moving from broad-spectrum inhibitors to isoform-specific modulators. Advances in selective PDE4 inhibitors, such as those targeting PDE4B and PDE4D, offer promising strategies for minimising systemic toxicity while enhancing therapeutic efficacy [211,213]. Additionally, the selective inhibition of specific PDE pools through AKAP-disrupting peptides, such as PI3Kγ mimetic peptides, presents another targeted approach for modulating compartmentalised cAMP signalling [176,215,216]. These innovations are particularly relevant for treating heart failure, where maladaptive remodelling and arrhythmogenesis demand precise modulation of cAMP signalling.

Future research should prioritise the development of isoform-specific therapies that will allow for more fine-tuning regulation of cAMP. Gene therapies, allosteric modulators and combination strategies targeting PDE4’s interactions with key proteins like AKAPs and POPDC1 can potentially transform clinical approaches. Furthermore, these therapies must be designed with an acute awareness of species differences to ensure successful translation from bench to bedside.

By bridging fundamental molecular insights with clinical applications, studying the PDE4 family opens pathways to novel treatments that restore cardiac function, mitigate heart failure progression and offer more targeted, safer therapeutic approaches.

## Figures and Tables

**Figure 1 cells-14-00460-f001:**
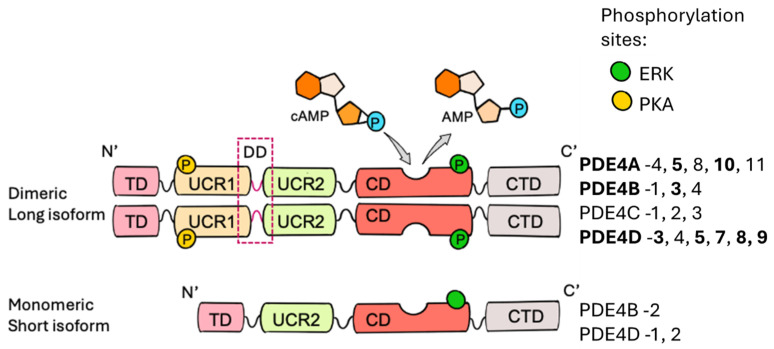
Structural organisation, classification, regulation and enzymatic activity of PDE4 long and short isoforms in the heart and cardiomyocytes. The diagram illustrates the domain architecture of dimeric long and monomeric short isoforms of PDE4. Long isoforms contain UCR1 and UCR2 domains, facilitating dimerisation via the dimerisation domain (DD, dashed box), a surrogate substrate of PKA phosphorylation, while short isoforms lack UCR1 and remain monomeric. The catalytic domain (CD, red) hydrolyses cAMP into AMP. Phosphorylation by ERK (green circle) and PKA (yellow circle) regulates PDE4 activity, affecting enzyme stability and function. The C-terminal domain (CTD) contributes to isoform-specific interactions within cells. Isoforms confirmed in cardiomyocytes are shown in bold.

**Figure 2 cells-14-00460-f002:**
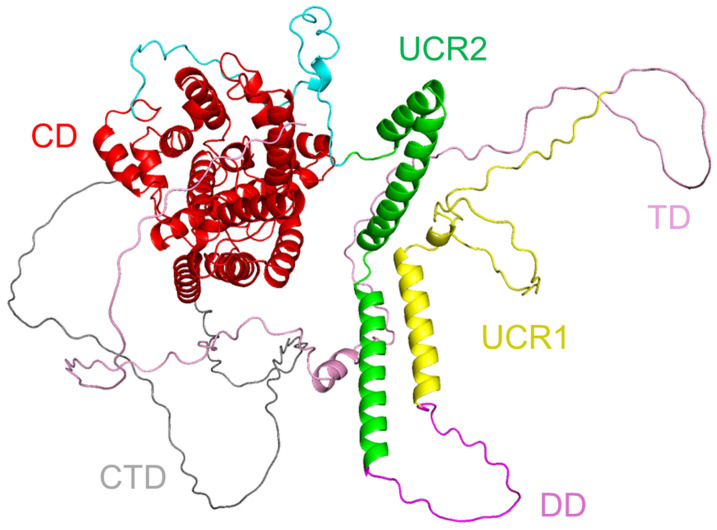
Three-dimensional structural model of human PDE4B protein. The structural organisation highlights key functional regions: catalytic domain (CD, red), C-terminal domain (CTD, grey), dimerisation domain (DD, magenta), targeting domain (TD, pink), upstream conserved region 1 (UCR1, yellow) and upstream conserved region 2 (UCR2, green). UniProt: Q07343; generated using AlphaFold 3 [88,89] protein structure database: AF-Q07343-F1-v4); domains were marked with PyMol 3.1.1 [90].

**Figure 3 cells-14-00460-f003:**
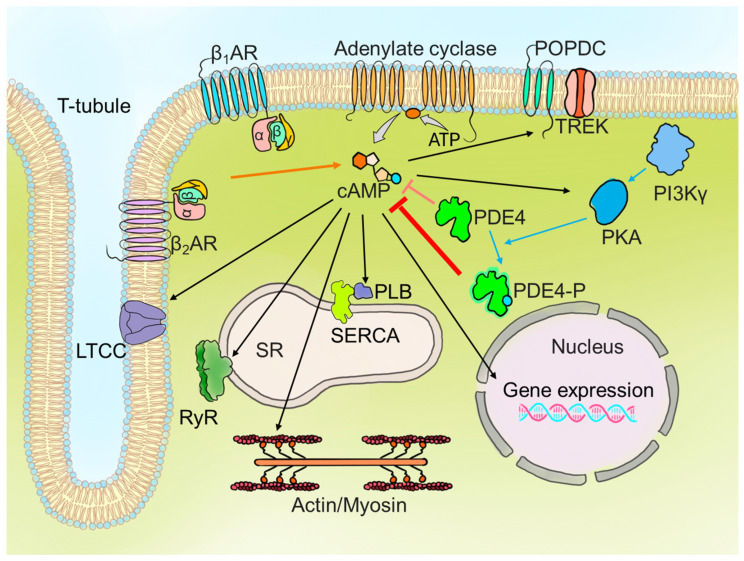
βAR regulation of cAMP signalling and PDE4 activity in cardiomyocytes. This schematic represents the βAR signalling cascade, which regulates cAMP levels and downstream targets in cardiomyocytes. Stimulation of β₁ARs and β₂ARs activates adenylate cyclase, converting ATP into cAMP. cAMP activates PKA, which phosphorylates various targets, including PLB and RyRs for calcium regulation, and LTCC to increase calcium influx. PDE4, regulated by AKAPs (particularly PI3Kγ), hydrolyses cAMP to AMP, modulating cAMP compartmentation and preventing excessive PKA activation. Phosphorylation of PDE4 further enhances its activity. PDE4 plays an important role in regulating nuclear gene expression and calcium handling, thereby regulating the contraction of cardiomyocytes.

**Table 1 cells-14-00460-t001:** Classification of phosphodiesterase families based on substrate specificity and isoforms.

Category	PDE Family	Isoforms	References
cAMP-specific PDEs	PDE4	PDE4A, PDE4B, PDE4C, PDE4D	[14,15]
	PDE7	PDE7A, PDE7B	[16,17]
	PDE8	PDE8A, PDE8B	[18,19,20]
cGMP-specific PDEs	PDE5	PDE5A	[21,22,23]
	PDE6	PDE6A, PDE6B, PDE6C	[24,25]
	PDE9	PDE9A	[26,27,28]
Dual-substrate PDEs	PDE1	PDE1A, PDE1B, PDE1C	[29,30,31]
	PDE2	PDE2A	[32]
	PDE3	PDE3A, PDE3B	[33,34,35,36]
	PDE10	PDE10A	[37,38]
	PDE11	PDE11A	[39,40]

**Table 2 cells-14-00460-t002:** Enzymatic properties and substrate specificity of cardiac PDEs.

Category	PDE Family	Main Functions and Localisation	Km (μmol/l)		References
			cAMP	cGMP	
cAMP-specific PDEs	PDE4	Plays a key role when cAMP levels are elevated. Detected in cardiomyocytes and fibroblasts.	1–6	NA	[15,47,48,49,50,51,52]
	PDE8	Controls ICa,L current. Detected in cardiomyocytes.	0.1–0.6	NA	[19,20,51,53,54]
cGMP-specific PDEs	PDE5	Preferentially regulates a pool of cGMP produced by soluble GC. Detected in cardiomyocytes and fibroblasts.	201	1–6	[2,21,22,23,51,55,56]
	PDE9	Preferentially regulates the NP-induced cGMP. Detected in cardiomyocytes and not detected in fibroblasts.	230	0.1–0.4	[2,26,27,28,51,57,58]
Dual-substrate PDEs	PDE1	Regulation of calcium/calmodulin. Detected in cardiomyocytes and fibroblasts.	1–125	1–8	[29,51,59,60]
	PDE2	Regulates local mitochondria-related cAMP pools. More abundantly expressed in cardiac fibroblasts compared to cardiomyocytes.	30–112	10–31	[6,51,61,62]
	PDE3	Responsible for the tonic effects in the myocardium. PDE3 is the most important in cardiomyocytes.	0.1–0.8	0.1–0.8	[33,34,36,60,63,64,65]
	PDE10	cAMP regulates PDE10 biphasically, modulating cGMP hydrolysis. Detected in cardiomyocytes and fibroblasts.	0.2–0.3	1.1–7.2	[38,45,51,66]

**Table 3 cells-14-00460-t003:** AKAP isoforms in cardiomyocyte regulation: key roles and references.

AKAP	Aliases	Functions in Cardiomyocytes
AKAP1	D-AKAP1 [139], AKAP121 [141], AKAP149 [142], S-AKAP84 [143], mitoAKAP [144]	Regulates mitochondrial dynamics, oxidative phosphorylation, and cardiomyocyte survival, playing a protective role against cardiac hypertrophy and heart failure [144,145].
AKAP2	D-AKAP2 [140]	Organises a signalling complex with PKA and Src3, promoting anti-apoptotic and pro-angiogenic responses essential for myocardial infarction recovery [146].
AKAP5	AKAP79 [147], AKAP150 [148], AKAP75 [149]	Coordinates PKA signalling in T-tubules and plasma membrane, regulating calcium channels and cardiac contractility under sympathetic stimulation [150,151].
AKAP6	mAKAPβ [152], AKAP100 [138], mAKAP [153]	Regulates calcium handling by interacting with PLN and organises the nuclear envelope microtubule-organisng centre through centrosomal and Golgi-associated proteins [154,155].
AKAP7	AKAP15 [156], AKAP18 [157]	Localises PKA to the plasma membrane, regulating membrane events like cardiac IK1 currents [158,159,160].
AKAP9	Yotiao [161], AKAP350 [162], AKAP450 [163]	Coordinates β-adrenergic regulation of the IKs potassium channel by assembling PKA, PP1, AC9, and PDE4D3 into a macromolecular complex. Disruptions are linked to long-QT syndrome and impaired cardiac repolarisation [164,165,166].
AKAP12	Gravin [167], SSeCKS [168], AKAP250 [169]	Mitigates maladaptive remodelling, oxidative stress, and fibrosis by inhibiting Ang-II-induced TGFβ1 signalling. Also regulates cardiac contractility and calcium handling during isoproterenol stimulation [53,170].
AKAP13	AKAP-Lbc [171]	Coordinates cardiomyocyte signalling pathways involved in protection against doxorubicin toxicity, pathological hypertrophy, and α1-adrenergic receptor-mediated RhoA activation [172,173,174].

**Table 4 cells-14-00460-t004:** PDE4 inhibitors: specificity, clinical indications and trial phases.

Drug Name	PDE4 Specificity	Disease	Phase	NCT Number *
Roflumilast [208]	Pan-PDE4	Polycystic Ovary Syndrome	IV	NCT02037672; NCT02187250
		Chronic Hand Eczema	IV	NCT05682859
		Ulcerative Colitis	IV	NCT05684484
		Chronic Obstructive Pulmonary Disease	IV	NCT01595750
Apremilast [209]	Pan-PDE4	Recurrent Aphthous Stomatitis (RAS)	IV	NCT03690544
		Alopecia Areata	IV	NCT05926882
		Chronic and Recurrent Erythema Nodosum Leprosum	IV	NCT04822909
		Oral Lichen Planus	IV	NCT06260904
Crisaborole [207]	Pan-PDE4	Moderate Atopic Dermatitis	IV	NCT04214197
		Seborrheic Dermatitis	IV	NCT03567980
Hemay005 [211]	Pan-PDE4	Behçet’s Disease	III	NCT06145893
		Severe Plaque Psoriasis	III	NCT04839328
Cilomilast [208]	Pan-PDE4, more selective for PDE4D	Chronic Obstructive Pulmonary Disease	III	NCT00103922
Tanimilast [211]	Pan-PDE4	Chronic Obstructive Pulmonary Disease and Chronic Bronchitis	III	NCT04636801
GSK256066 [210]	PDE4B	Chronic Obstructive Pulmonary Disease	II	NCT00549679
BPN14770 [205]	PDE4D	Fragile X Syndrome	II	NCT03569631
KIT2014 [176]	PDE4B and PDE4D	Healthy Subjects	I	NCT06659757

* ClinicalTrials.gov identifiers from https://clinicaltrials.gov (accessed on 16 March 2025).

## Data Availability

Not applicable.

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
