# Peer review of "Role of PDE4 Family in Cardiomyocyte Physiology and Heart Failure"

_cells, 2025, doi:10.3390/cells14060460_

Round 1
Reviewer 1 Report
Comments and Suggestions for Authors
This review manuscript is very well crafted to summarize the findings of PDE4 in cardiac biology. It is an easy read with very helpful tables and figure. Furthermore, the amount of studies included in this review should be of great assistance for a big crowd of readers in the future. With that, this manuscript can be further improved in a few areas that are listed below.
- There is a section specifically on PDE4A (2.4), which is a little bit odd as it is the only one out of all PDE4 isoforms. That may be strengthened by including section with table on the currently available genetic mouse models and their CV phenotypes.
- In section 4.1, the PKA-R1 and PKA-RII compartments are not clearly defined and explained.
- Lastly, authors should include a table of currently available PDE4 inhibitors (in trials or in research use) and describe the sub-isoform selectivity, including references. This manuscript can then be one of the most resourceful place of PDE4 related studies.
Author Response
Please see the attachment. All point-by-point responses to the reviewers' comments have been provided in the attached Word file, formatted according to the template

Reviewer 2 Report
Comments and Suggestions for Authors
Review of Cells 3505116
Role of PDE4 family in cardiomyocyte physiology and heart failure by I Sherstnev et al
This is an excellent in depth review of the role of PDE4 in cardiac physiology and pathophysiology. The authors make the case that although PDE4 only accounts for about 10% of the total PDE activity in the human heart, the realization that the PDE4 isoforms, PDE4A, PDE4B and PDE4D, provide compartmentalized activity in a variety of subcellular nanodomains, suggests they may largely be responsible for spatial and temporal precision of cAMP signaling in the heart, especially in response to ß-adrenergic stimulation, and thus may provide excellent therapeutic targets for a host of cardiac pathophysiologic conditions. The authors provide an extensive review of the structure of the expressed isoform members of the PDE4 gene family, their expression and function in cardiac tissues across different species, their roles in modulation of ß-adrenergic signaling and calcium regulation, and what is currently known about the development of inhibitors of PDE4, both competitive and allosteric, and PDE4 degraders, such as PROTACS. Inasmuch as the compartmentalization of PDE4 in subcellular nanodomains is achieved through binding into complexes with AKAPs, the authors go fairly heavily into depth about AKAPS, listing the 13 types known and their functions in relation to cardiac physiology. Overall I think this review, with its focus on the role of PDE4 in cardiac function, is extremely well done and provides a contribution to the literature of an important and specific area that has not been very extensively covered prior to this. I list a few very minor corrections that are needed and provide some comments and suggestions on a few things that the authors can implement to further enhance this manuscript, as follows:
- In the Abstract lines 18-21, the authors state: “ Unlike other PDEs, PDE4 provides spatial and temporal precision to cAMP signalling, particularly under β-adrenergic stimulation, through its compartmentalised activity in subcellular nanodomains, including the sarcoplasmic reticulum, plasma membrane, and nuclear envelope.” Inasmuch as it has been shown recently that PDE8A is compartmentalized in heart through binding to AKAP12 (PMID 38506047) and it is believed that other PDEs in heart may also be compartmentalized, I think it would be best if the authors left out the beginning phrase of that sentence: “Unlike other PDEs…” and just started the sentence with “PDE4 provides spatial and temporal….” etc.
- Another review article, published in another MDPI journal in 2023, focuses on PDE4 in cardiovascular diseases and parallels some of what is covered here (PMID 38069339). I think it would be appropriate for the authors to cite this review to bring it to the attention of the reader somewhere in their Introduction.
- On lines 60-61 it states “In the heart, seven PDE families have been identified: PDE1 [39], PDE2 [30], PDE3 [33], PDE4 [40], PDE5 [41], PDE8 [17] and PDE9 [42], nevertheless…”. I would change that to eight PDE families and add PDE10, as PDE10 has also been shown to be expressed in heart tissue where it is upregulated in heart failure and plays a role in pathological cardiac remodeling and dysfunction
(PMID 31801360).
- Although it is understood that the authors wish to concentrate on PDE4 and focus on its functions in the heart, inasmuch as the other PDEs also have important functions in cardiac physiology, I think it would be useful for the authors to add an additional table listing all the PDEs known to be expressed in cardiac tissue with a statement(s) in the table indicating their function in heart, where known, so as to highlight how the functions of PDE4 might differ from these. Much of this information can found in recent reviews on PDEs in heart by Kim and Kass (PMID 27787716) and Kamel Leroy, Vandecasteele and Fischmeister (PMID 36050457), as well as in other papers.
- Also given that there are as many as eight PDE gene families expressed at the protein level in heart tissue, it would be useful, where known, for the authors to indicate which PDEs are actually expressed in cardiac myocytes, rather than coming from other cell types, like endothelial cells, fibroblasts, or blood cells in blood vessels of the heart tissue.
- There are a number of places throughout this manuscript where citations are given as the author and year published, rather than a number referring to the list of references at the end. For example, lines 146, 183,191, 226, 230, and 588. This needs to be fixed.
- In section 2.1, in the paragraph beginning on line 143, the authors make the case that since the Km of PDE3 is about 10 fold lower than that of PDE4, they say that “This suggests that PDE3 primarily regulates cAMP levels under basal conditions, while PDE4 plays a more prominent role when cAMP concentrations rise, such as during β-adrenergic stimulation [76, 77].” It should be noted that cardiac tissue expresses appreciable amounts of PDE8A, as well as PDE8B (eg - PMID 39270009, PMID 38603476, PMID 38506047, PMID 36810794, PMID 20353794), and inasmuch as PDE8 has a Km 140 times lower than that of PDE4, PDE8 is primarily thought to regulate cAMP levels under basal conditions. The authors should modify what they say here to take this into consideration.
- In line 277 it is stated “In neonatal mouse cardiomyocytes, PDE4D demonstrates isoform-specific regulation of ß2AR signalling. Notably PDE4D selectively regulates β2AR without affecting β1AR signalling, emphasising its role in localising and fine-tuning cAMP signals [100].” And on line 299 when discussing a paper on PDE4 regulation of calcium dynamics during ß-adrenergic stimulation it states “They also identified that PDE4 activity diminishes with advancing age and experiences a further decline in patients with AF, thereby linking PDE4 dysfunction to increased arrhythmic vulnerability.” This indicates that changes in cardiac PDE4 expression during development and aging is important to be aware of and to address. Complementing what the authors state here, there was a paper published in Biochem J. in 1987 on Ontogenetic Changes in Adenylate Cyclase, cAMP Phosphodiesterase and Calmodulin in Chick Ventricular Myocardium (PMID 2820384). Although this paper was published prior to the current known nomenclature for PDEs, the paper showed that similar to human heart which we now know abundantly expresses PDE1C (PMID 17726023), the major PDE activity in supernatants of chick ventricles is also constituted by a Ca2+-calmodulin sensitive PDE with Kms for both cAMP and cGMP of 1-2 µM (ie. -PDE1C). This was actually one of the first evidences of a Ca2+-calmodulin sensitive PDE with high affinity for both cAMP and cGMP, distinguishing it from PDE1A and 1B, which have much higher Kms for cAMP, as PDE1C had not been discovered yet. Chicks hatch 21 days after fertilization and PDEs were examined and isolated by DEAE-Sephacel chromatography, sensitivity to Ca2+-calmodulin, and kinetics for hydrolysis of cAMP and CGMP at 8 day embryonic (8E), 18 day embryonic (18E), 6 day hatched (6H), and adult Chick ventricular myocardium. Results showed a single major peak of PDE activity eluting from DEAE-Sephacel in 8E, 6H and 6 month adult chick hearts with all the characteristics consistent with PDE1C. However supernatants from the late embryonic 18E ventricles clearly showed two equivalent peaks of PDE activity, with the peak eluting at lower salt concentration consistent with PDE1C, and the peak eluting at the higher salt concentration consistent with PDE4, based on substrate specificity, kinetics, and elution profile on DEAE. This is noteworthy because it had been known that there is 10-fold sub-sensitivity to ß-adrenergic stimulation by isoprenaline of the force of cardiac contraction in late embryonic (16E-21 E) ventricles as compared to earlier embryos and hatched chicks (PMID 61095800). Interestingly, 16E is the day when adrenergic neuroeffector transmission is first observed in the chick embryo right ventricle suggesting that the PDE4 expressed at this late embryonic stage may function to limit ß-adrenergic stimulation at that developmental stage as some sort of protective mechanism associated with the beginning of adrenergic nerve innervation to the right ventricle occurring at that time. It might be useful for the authors to mention something about this along with their comments on neonatal mouse cardiomyocytes.
- On line 469, where it is mentioned that AKAP12 anchors PDED3 and PKA to subplasmalemmal domains, it would be advisable to mention that AKAP12 also anchors PDE8A as well (PMID 38603476, PMID 38506047).
- In section 4.2 in the discussion of AKAPs in Heart Failure, I think it would be better to use the nomenclature presented in Table 2 when referring to specific AKAPs. For example in line 533 replace AKAP121 with AKAP1, line 539 replace AKAP-Lbc with AKAP13, and line 543 replace mAKAPß with AKAP6.
Author Response
Please see the attachment. All point-by-point responses to the reviewers' comments have been provided in the attached Word file, formatted according to the template.
